# Determining hydrological flow paths to enhance restoration in impaired mangrove wetlands

**Rosela Pérez-Ceballos[1], Arturo Zaldívar-Jiménez [2]\*, Julio Canales-Delgadillo[1], Haydée López-Adame[2], Jorge López-Portillo[3], Martín Merino-Ibarra[4]**

**1** CONACYT Instituto de Ciencias del Mar y Limnología Estación El Carmen UNAM, Cd. del Carmen, Campeche, México, **2** ATEC Asesoría Técnica y Estudios Costeros SCP, Mérida, Yucatán, México, **3** Instituto de Ecología A.C., Xalapa, Veracruz, México, **4** Unidad Académica de Ecología y Biodiversidad Acuática, Instituto de Ciencias del Mar y Limnología, Universidad Nacional Autónoma de México, Circuito exterior S/N, Ciudad Universitaria, Coyoacán, México

\* arturo.zaldivar@atecscp.com

**Data Availability Statement:** The data underlying this study have been uploaded to figshare. Data are available from the following links: https://doi.org/

## Abstract

The restoration of mangroves has gained prominence in recent decades. Hydrological rehabilitation has been undertaken to connect impaired mangroves with the sea, lagoons or estuaries. Because mangrove hydrodynamics occurs on the surface and interstitial spaces in the sediment, we propose to determine the hydrological flow paths to restore the hydrological regimes of the impaired mangroves. The hydrological flow paths were determined through a micro basin analysis based on microtopographic data to generate a digital elevation model. Applying this methodological approach, the hydrology of an impaired area on a barrier island in the Gulf of Mexico was restored by excavating, desilting or clearing the channels on the identified hydrological flow paths. This area was compared to one in which impaired mangroves were reconnected to the marine lagoon but disregarding the flow paths. Data collected in both areas were evaluated by flood level analysis, using two methods: (i) a simple linear regression model (SLRM) and (ii) spectral analysis (SA), also known as dominant frequency analysis. The results suggest that restoration based on the hydrological flow paths was more effective than the direct opening to the nearest main water body without accounting for the microtopography. In both areas, soil salinity and sulfides decreased after hydrological reconnection. However, a greater efficiency in the investment of time and human resources was achieved when preferential flow paths were identified and taken into account. The methodological procedures described in this study are of universal application to other mangrove restoration programs.

## Introduction

Mangroves are found at the land-sea interphase in the tropical and subtropical regions of the world [1,2]. These systems protect coastlines against erosive and flooding processes caused by storms and hurricanes [3] and improve coastal water quality [4], among other

10.6084/m9.figshare.11496675; https://doi.org/10.
6084/m9.figshare.11497044; https://doi.org/10.
6084/m9.figshare.11497050; https://doi.org/10.
6084/m9.figshare.11497071; https://doi.org/10.
6084/m9.figshare.11497143; https://doi.org/10.
6084/m9.figshare.11497158; https://doi.org/10.
6084/m9.figshare.11497074; https://doi.org/10.
6084/m9.figshare.11497077; https://doi.org/10.
6084/m9.figshare.11497095.

**Funding:** The authors received no specific funding for this work.

**Competing interests:** The authors have declared that no competing interests exist.

environmental services. Their horizontal distribution is restricted to the intertidal zone, and they are hydrologically and biologically connected to other water bodies or marine ecosystems such as tropical flooded forests, marshes, seagrasses and reefs. Mangroves form a habitat for diverse vertebrate and invertebrate groups [5] and are among the most efficient ecosystems to the sequestration of carbon [6,7]. However, they are also among the most threatened ecosystems worldwide [8].

Different scenarios of sea level rise predict the possibility of more frequent flooding events that may impair areas of mangrove vegetation due to the alteration of the hydrology, with severe consequences for the survival, establishment and regeneration of this ecosystem [9]. Sea level rise has a greater effect on mangroves from carbonate environments', since their adequate functioning depends on the relative elevation of the soil regarding to the flooding level [10]. Moreover, tropical storms and road infrastructure construction frequently have an immediate and severe effect on the hydrological flow regimens in mangrove areas [11]. Thus, it is essential to develop strategies against the effects of global climate change and for the recovery of ecological processes through ecological restoration [12]. Therefore, it is necessary to monitor key environmental factors, such as hydrology and biogeochemical characteristics, to use them as mangroves restoration success indicators [13].

The hydrology of mangroves has three components: the hydroperiod, the hydrodynamics and the water inputs given by rainfall, surface and subsurface fresh or seawater [14]. The hydroperiod is the most important one, characterized by the level, frequency and duration of flooding through time [15–17]. The hydroperiod is the hydrological signature of each mangrove ecotype [18] and determines the gradients of nutrients (such as N, P, and C) and biogeochemical regulators of the soil (salinity, sulfides and oxidation-reduction). The water flow into the mangrove results from the seasonal regimes of the water sources, such as the astronomical and meteorological tides [19]. The currents of water flowing out of the mangrove are stronger than the currents flowingin, which occurs mainly during high or extraordinary tides and defines the regimes of water circulation and velocity within the mangrove. The water flows into the mangrove through the surface and through the subsurface (porewater) although it is lower due to sediment resistance [20,21]. The hydrological connectivity in mangroves is regulated by their interaction with the adjacent water bodies through lotic, semi-lotic or lentic currents and by tidal exchange [20,22]. For this reason, a method for quantifying hydrological connectivity in mangroves is to measure the duration of flooding (i.e., hours per month) [21,23].

The mangrove environment can vary between extreme periods of minimum and maximum flooding. Low tides and scarce precipitation characterize the minimum flooding period, which produces a heterogeneous soil matrix and controls the biogeochemical characteristics by increasing the salinity as a result of a higher evaporation rate and low connectivity. In contrast, high tides and abundant precipitation characterize the maximum flooding period which produces a homogeneous soil matrix, lower salinity and high connectivity [24].

Mangrove degradation begins with the partial or total interruption of the hydrological connectivity due to anthropic or natural disturbances that drastically modify the hydroperiod [25–27]. Because of a reduced water flow, deposition of sediments in the tidal channels increases [28], along with the duration of the flood. The evaporation in turn, increases the concentration of salts and sulfides in the porewater to the point that these elements become stressors that hinder seedling establishment and adult survival [16,29]. In order to reverse this process of deterioration, the first necessary action is to recover the hydrological, biogeochemical and biological condition of the mangroves by implementing hydrological and sedimentological restoration [13,26,30]. Unfortunately, it is difficult to determine the exact location in which to conduct such a restoration. This is because the topographic conditions of the mangroves may

complicate the identification of the tidal channels that should connect to the adjacent water bodies. Moreover, the impaired mangrove soils undergo processes of reduction and oxidation that depend on the presence or absence of standing water [31] and that favor soil subsidence due to the increased decomposition rate of dead roots. These factors further complicate the identification of hydrological flow paths.

Some proposed using hydraulic modeling to simulate flooding to perform hydrological and sedimentological restoration [32], while others considered the presence of mangrove species to classify the hydrological patterns and to make decisions on restoration actions [33]. Moreover, strategies aimed to improve mangrove recruitment have been applied. However, as it requires the modification of the terrain's relief [34], the necessary investment is always considerable. A less costly method is to connect the impaired mangroves with the adjacent main water bodies to re-establish hydrological flows [27,35,36].

We used hydrological restoration to improve water and soil quality with the aim to promote the natural regeneration of the vegetation cover and to recover the ecosystem functions of impaired mangroves [27,35–39]. Considering the diversity of existing restoration strategies [40], the objective of this study is to describe the application of microtopographic analysis to identify the hydrological flow paths to improve the results of hydrological and sedimentological restoration in an impaired mangrove forest. The measurement and characterization of the hydroperiod, flooding patterns and biogeochemistry characteristics as salinity and sulfide concentration in porewater were considered as indicators of restoration success. Finally, the monetary expenses of this strategy were compared with those of a different methodological approach implemented in a nearby area, where the impaired mangrove was directly connected to the adjacent water body without taking into account the preferential hydrological flow paths.

We hypothesize that restoration would result in more dynamic reconnection and water flow following the preferential flow paths that merely connect the impaired mangroves to many tidal channels along the perimeter of Laguna de Términos. By modeling the preferential flow paths, we expect that natural regeneration of impaired mangroves will occur in a shorter time and that the monetary investment will be lower.

## Materials and methods

### Study area

The study site (18.672544˚N; 91.667494˚W) is located in Isla del Carmen Campeche, Mexico, a barrier island formed by carbonate sediments. The island is bordered by Laguna de Términos in the south and by the Gulf of Mexico in the north (Fig 1A). There are four tributary rivers that discharge continental water, sediments and nutrients into the lagoon. The climate is warm to humid with a yearly average temperature of 34 ˚C and 1,680 millimeter (mm) of annual rainfall concentrated between June and October [41].

The dominant winds blow in the NNW and ESE directions [42,43]. The daily tide regimes have an average of about 0.43 m of amplitude. Because rainfall is scarce during the dry season (April to May), there is no flooding, and the minimum tides are about -0.24 m [43]. In contrast, during the late autumn and winter (November to January, also locally called "nortes" season), moderate rainfall, the lowest temperatures, and strong northerly winds are present, there is regular flooding, and the tide reaches its maximum interval of 0.92 m [43]. The salinity of the lagoon ranges from 18 to 36 [44]. The vegetation around the lagoon includes freshwater marsh vegetation, mangroves, and coastal dunes [45].

Mangrove forests dominate Isla del Carmen and are distributed according to topography and tidal influence [46]. The soil in the mangroves have about 0.71% nitrogen, 0.047%

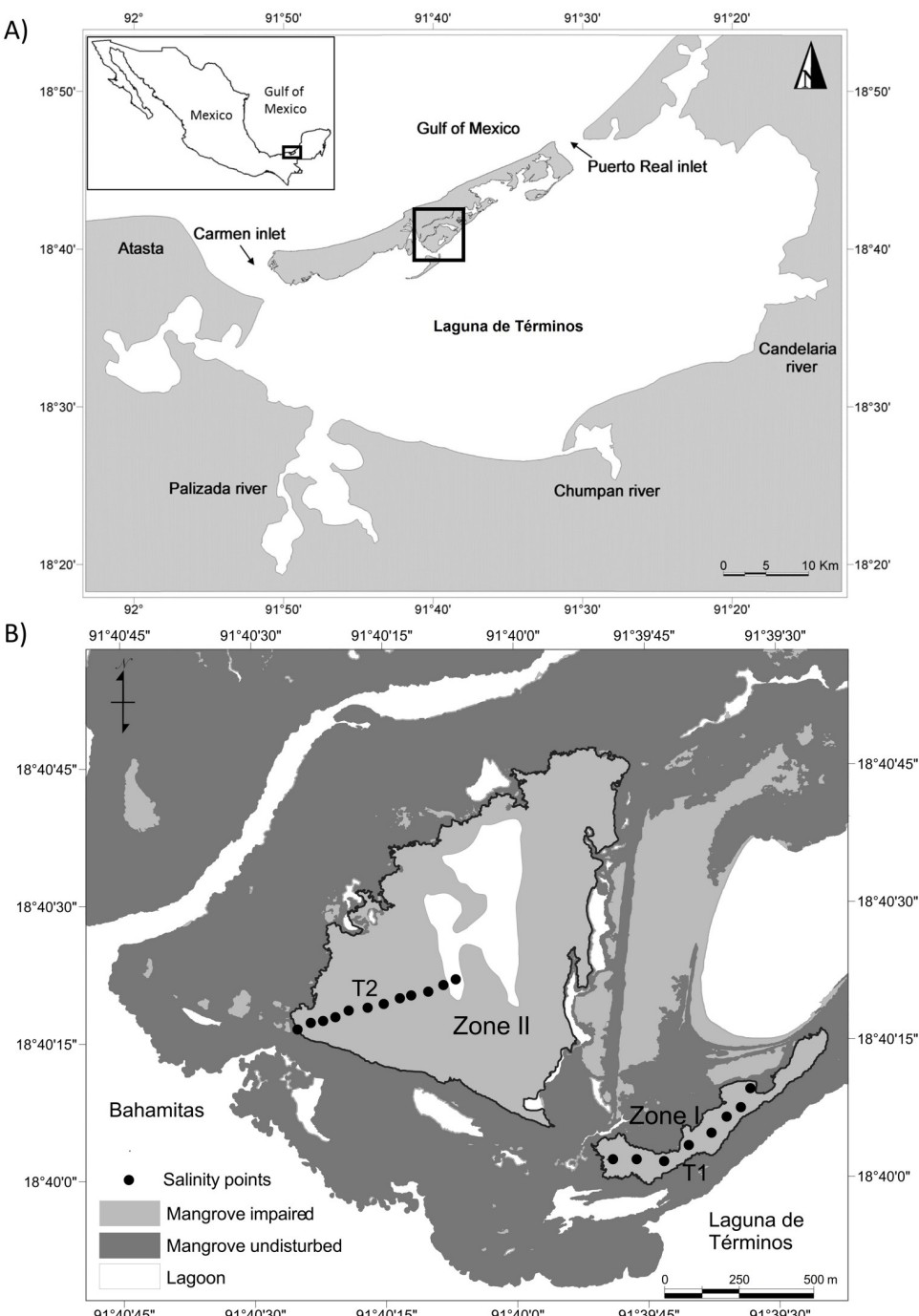

**Fig 1.** (A) Location of Isla del Carmen, a sandy barrier delimited to the north by the Gulf of Mexico and to the south by Laguna de Términos. (B) Zones I and II where hydrological restoration was carried out. The undisturbed (dark gray) and impaired (light gray) mangroves are shown. The sampling stations (black circles) where porewater salinity was monitored are distributed along two transects (T1-T2).

phosphorus, and 16–30% organic matter, mainly of autochthonous mangrove-derived peat with a mean bulk density of 0.29 g cm$^{-3}$ [46,47]. Two main mangrove communities are present in the study area: fringe forests dominated by red mangrove (*Rhizophora mangle L.*) and a basin forests dominated by white mangrove (*Avicennia germinans L.*) [48]. The fringe forests

are more frequently flooded by tides than the basin forests, which are more vulnerable to over-flooding and draining [48]. In the study area, the death of mangrove trees in the basin forests occurred when trees fell on the tidal channels after two severe tropical storms struck the island in 1995. The fallen trees reduced the flow of water to the basin mangroves and increased channel sedimentation, which over the years led to impaired mangroves. We selected two zones with a high proportion of impaired mangroves (Fig 1B): Zone I, an area of 19 ha bordered on the north side by a basin forest of 9 ha of monospecific *A. germinans* of about 4–5 m in height, while on the south side, there is a 5 ha thin slide (900 m x 55 m) of the same type of mangrove forest. Zone II is an area of 75.5 ha bordered by a patch of basin forest of about 86 ha of *A. germinans* at 6–7 m in height, and 16 ha of a fringe mangrove of *R. mangle* at 8–9 m in height on the southwest side. The access to the study area was permitted by the Secretaría de Medio Ambiente y Recursos Naturales (SEMARNAT) under the authorization number SEMARTAT/IACC/0394/2010.

## Hydrological restoration

Zone I was intervened from November 2010 to December 2012 and Zone II was intervened from October 2011 to January 2013. In both zones, the hydrological restoration included the cleaning of the main tidal channels by removing the fallen trees (which were up to 20 cm in diameter at breast height (DBH)) that obstructed the ebb and flow of water. In Zone II, it was also necessary to excavate secondary channels by moving the sediments out of the flow paths to allow for the movement of water to and from impaired mangroves.

**Microtopography and determination of the preferential flow paths.** In mangroves, the microtopography of hydric soils strongly controls the flow of water, forcing its convergence or divergence in response to height differences of a few millimeter. To characterize the microtopography of the terrain (Zone II), we used the stop-and-go mode with readings from a DGPS method at a single known point (18.651603 N, 91.7591042 W and 0.446 masl referenced to WGS84) [49]. With the collected data, a digital elevation model (DEM) was generated and processed by using the watershed analysis included in the surface analysis module of the Microimages software (TNT Mips ver. 2013, Nebraska USA). This procedure was used to identify preferential flow paths along microtopographic gradients to connect Zone II to the adjacent water body (Fig 2A) [50].

**Excavation of channels.** The restoration of the main tidal channel in Zone I included the excavation of 824 m in length, 3 m in width and 0.8 m in depth to connect it by two inlets to the south of Laguna de Términos. Seven secondary channels of about 2 m wide and 0.7 m in-depth and a total length of approximately 1,689 m were excavated in a zig-zag shape to increase the lateral influence area of the channel (i.e., the hydrologic signature).

In Zone II, a natural tidal channel of 1,730 m in length (thick line in Fig 2B) at the east side of the area was dug 1–2 m deep and 3–5 m wide to eliminate obstacles to water flow. Following the identified preferential flow paths, nine secondary channels (about 5,313 m) were connected to the natural tidal channel. Although it was not suggested as a preferential flow path, an additional secondary channel was dug at the west side of Zone II to connect it with the Laguna de Términos (Fig 2B). This action carried out to induce variations and responses in the system regardless of the preferential flow results.

## Environmental monitoring

To collect data on the hydrological and soil characteristics, one sampling site was established in Zone I (site 4) and four were established in Zone II (sites 1, 2, 3, and R; Fig 2B). The site R is a reference site. It locates in the natural mangrove forest adjacent to the restoration site. It has

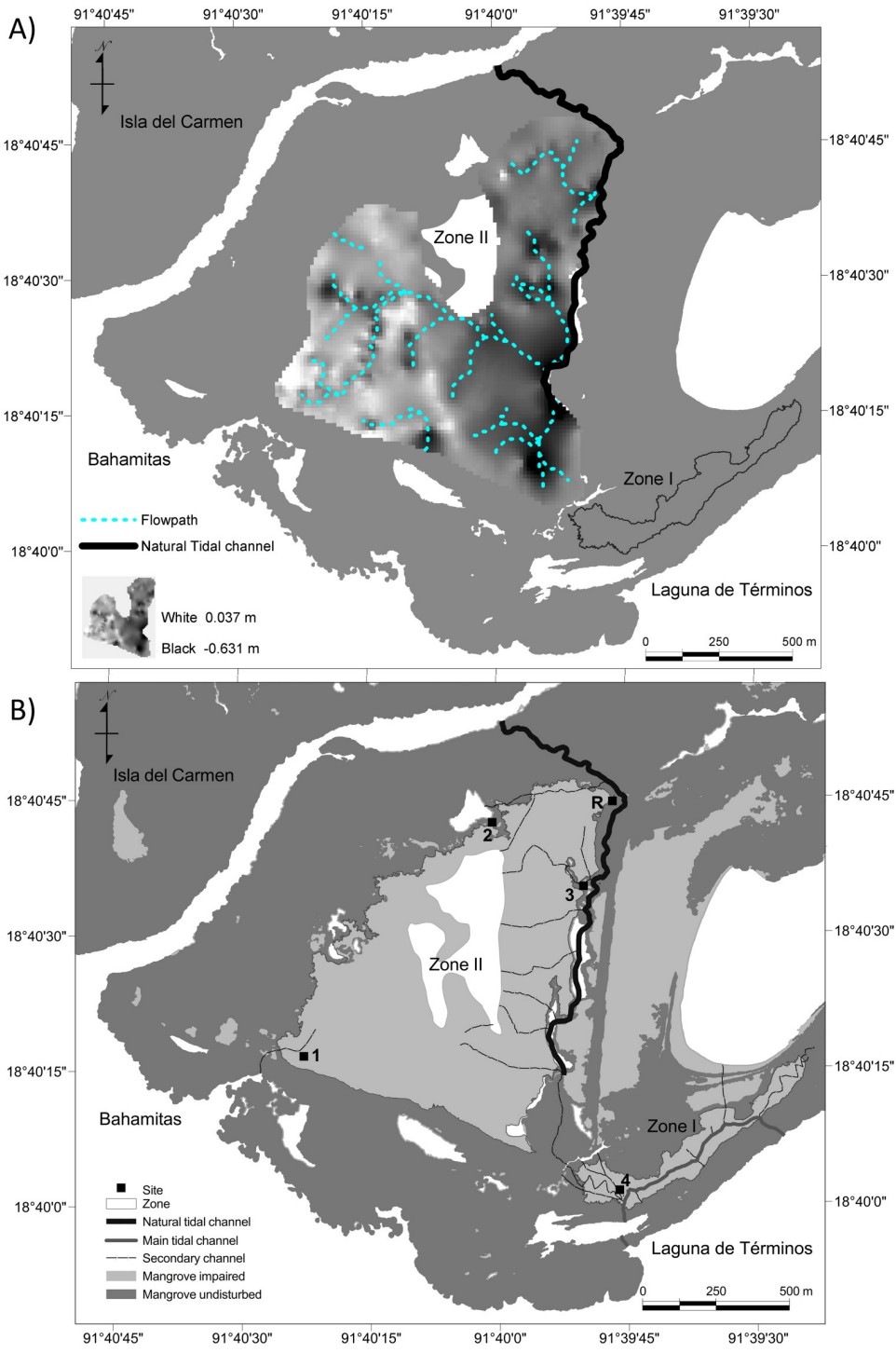

**Fig 2.** (A) Superposition of the microtopography of 75.5 ha as a tridimensional image in Zone II. The dotted lines indicate the dendritic pattern of the preferential flow paths that converge in the lowest points of the undisturbed mangrove. (B) Actual excavated main channels (thin lines) based on the preferential flow paths derived from the microtopography. The thick solid line indicates the main tidal channel that links the internal water bodies with Laguna de Términos.

a forest structure characteristic of *R. mangle* and *A. germinans*, with trees of 5.85 m in height and density of about 3,400 ind ha$^{-1}$ [51].

Prior to the restoration (2010), *A. germinans* was the dominant species in all sites as indicated by dead tree trunks and relict mangroves at sites 2 and 3, but only dead trunks at sites 1 and 4. At the end of the study (2014), there were seedlings and saplings of *R. mangle* and *A. germinans* at site 1 (mean height = 0.43 m; density = 111 ind ha$^{-1}$); there were surviving and developing shrubs of *A. germinans* at site 2 (1.86 m; 1,450 ind ha$^{-1}$); slightly taller trees of *A. germinans* at site 3 (2.3 m; 1,200 ind ha$^{-1}$); and *A. germinans* saplings at site 4 (0.55 m; 82 ind ha$^{-1}$).

**Water level.** At each site, a HOBO water level logger (U-20 ONSET USA) was installed to record changes in frequency, time, and flood level to identify seasonal patterns, as well as short term variations of the tides during the years 2012 (at the time of restoration), 2013, and 2014 (after restoration, Fig 2B). The sensors were located below soil surface level (at least 40 cm), attached to a string fixed to the top of a two-meter long PVC tube buried to a depth of 0.80 m and grooved every five cm. The water level loggers collected data every 60 minutes. All sites were georeferred using a single known point (18.651603 N, 91.7591042 W and 0.446 masl referenced to WGS84) [49].

**Biogeochemistry.** The biogeochemical characterization was carried out using two transects of 600 m in length (Zone I = T1, Zone II = T2, Fig 1B). Along each transect, ten sampling stations were established at a distance of 50 m from each other. Samples were collected during and after the restoration (Zone I: 2010–2014, Zone II: 2012–2014) at a depth of 35 cm using a 60 ml syringe coupled to a plastic hosepipe [52]. An aliquot of each sample was used to determine the temperature and salinity with a portable YSI 30 meter (YSI Inc., Yellow Springs, Ohio, USA).

In addition, the effect of the flooding periods (minimum and maximum) on the biogeochemical conditions; sulfide, and salinity concentrations were measured during the years 2013 and 2014 in the undisturbed mangrove (site R), and in the impaired sites subject to restoration (1 to 4). An impaired mangrove with dead trees of *A. germinans*, but with no restoration activities, was added to compare sulfide concentrations to those of site R. The concentration sulfide was measured by the methylene blue method [53] using a V-2000 field spectrophotometer (CHEMetrics Inc., Midland, VA).

## Data analysis

**Analysis of the water level.** The comparison between sites was carried out in two ways: (i) through a simple linear regression model (SLRM) and (ii) using spectral analysis (SA) or dominant frequency analysis applied to the time series of the water level. The SLRM considered the flood time recorded in reference site R as the response variable and the flood time recorded in the impaired sites subject to restoration (1 to 4) as explanatory variables. To analyze the influence of the rainfall on the hydrological connectivity, an additional multiple linear regression model (MLRM) was carried out for each site, but including rainfall (mm month$^{-1}$) data for all years. The hypothesis to be tested is that there is a significant linear relationship when comparing sites 1–4 and site R, and that the correlation coefficient ($r$) will increase as the flood patterns become more similar after hydrological reconnection.

The hydrological processes characteristic of the Laguna de Términos are the astronomical tide, the water discharges, and the oceanic circulation, which modulate the water levels in a way that differs from that of climatic season. Thus, if the hydrological and sedimentological restorations are successful, the dominant frequencies will be similar between site R and the restored sites (1–4), indicating synchronized seasonal and tidal (astronomical or meteorological) variations [42].

The time series of the flood levels allowed distinguishing the seasonal variation (monthly, semesterly and annually), as well as the tidal changes in specific cycles. The change of the water levels is a physical process that can be analyzed using sinusoidal functions of a known frequency, e.i., the sampling interval. A signal recorded within the time domain can be transformed into dominant frequency or Fourier's spectral analysis [19,54,55]. The dominant frequencies are sinusoidal functions of known frequency in the sampling interval, and, therefore, the Fourier transform is a discrete-time series $x(n)$ of finite length $N$ (number of data), obtained at a uniform sampling frequency f(s)

$$X(k) \sum_{n=0}^{N-1} x(n)e^{-2\pi kn/N} \quad k = 0, 1, \ldots, N-1 \tag{1}$$

where X(k) is the discrete Fourier series. For a geometric interpretation, the discrete Fourier transform:

$$X(k) = \sum_{n=0}^{N-1} x(n)\left( cosine\frac{2\pi kn}{N} - i \ sine \ \frac{2\pi kn}{N} \right) \qquad k = 0, 1\ldots, N-1 \tag{2}$$

The discrete Fourier series X(k) is a complex series of the same length (N) as the original time series x(n), which comprises a part of the real cosine a(k) and a part of the imaginary sine b(k):

$$X(k) = a(k) - ib(k) \qquad k = 0, 1\ldots\ldots, N-1 \tag{3}$$

where:

$$X(k) = \sum_{n=0}^{N-1} x(n)cosine\frac{2\pi kn}{N} \tag{4}$$

and

$$X(k) = \sum_{n=0}^{N-1} x(n)sine\frac{2\pi kn}{N} \tag{5}$$

The length of series (N) and sampling frequency f(s) are used to determine the $K^{th}$ Fourier coefficient:

$$f_k = kf_s/N \tag{6}$$

The Fourier transform allows representing a time series by a series of cosines and sines whose frequencies are multiples of $f_s/N$.

The SA was divided into three stages that correspond to each assessed year: 2012 (during), 2013 and 2014 (after) the restoration activities occurred. Once the SA was performed, a spectrum of dominant frequencies was plotted for sites 1 and R.

**Statistical analyses.**   To test for significant differences in the frequency of inundation, water level, and flooding duration variables during as well as after the restoration, we applied Wilcoxon and Kruskal-Wallis rank sum tests since the data did not meet the assumptions of normality and homocedasticity. The analyses were conducted for 2012, 2013, and 2014 in the five sites (1 to 4 and R). For each site, the hydroperiod variables were compared between years.

When significant differences in these variables were detected between the treatments, Tukey-Kramer pairwise comparisons were performed using the PMCMR routine of the statistical software R [56].

The exploration of the salinity data included Shapiro-Wilks tests and Levene's or Bartlett's test to check the assumptions of normality and homocedasticity. The Shapiro-Wilks test estimates the variance of the samples as a regression line in a quantile-quantile plot and as an estimator of the population variance. If the quotient of both estimated values is close to 1, then the samples come from a normal distribution. The Levene's test is useful to compare the equality of the variances between groups of samples using the group medians, while Bartlett's test is based on a $\chi^2$ statistic with $k^{-1}$ degrees of freedom, k being the number of categories or groups in the data set. Salinity values were centered using the R function scale(), while sulfide data were normalized to get approximately normally distributed data before the analyses.

Comparisons of salinity values during and after the restoration activities were carried out using a two-sided t-test. All of these tests were performed using the "car" and "stat" R-packages [57,58]. To explore the effects of restoration in salinity and sulfide concentration for the minimum and maximum flooding periods, the reference and restoration sites were compared between them and with an external impaired site through two-way analysis of variance.

## Results

### Tidal channel restoration

In the impaired mangrove of Zone II, which lack vegetation, the microtopographic gradient ranged from -0.63 to 0037 masl. The microtopographic gradient of the sites with relict vegetation ranged from 0.12 and 0.14 masl in sites 2 and 3, respectively. Meanwhile, the microtopographic level on the reference site averaged 0.19 m. In the area with no natural mangrove forest, elevation points of 0.037 m were found to the west and north sides of Zone II, which was close to undisturbed mangrove vegetation.

### Hydroperiod

Prior to restoration (2012), There were no significant differences in the frequency of inundation between sites in 2012 (Kruskal-Wallis test, $\chi^2 = 3.67$, df = 4, p = 0.451). Regarding the flooding duration as well as the water level, there were statistically significant differences between sites in this year ($\chi^2 = 37.84$, df = 4, p < 0.0001; $\chi^2 = 34.95$, df = 4, p < 0.0001, respectively). For 2013, significant differences between sites were observed regarding the frequency ($\chi^2 = 21.93$, df = 4, p < 0.0002), the duration ($\chi^2 = 23.42$, df = 4, p = 0.0001), and the level of flooding ($\chi^2 = 15.45$, df = 4, p < 0.0038). Similarly, in 2014 the frequency ($\chi^2 = 32.79$, df = 4, p < 0.0001), the flooding duration ($\chi^2 = 28.12$, df = 4, p = 0.0001), and the water level ($\chi^2 = 18.63$, df = 4, p < 0.0009) differed significantly between sites (Fig 3).

The regression coefficients between the flooding duration in sites 1, 2, 3 and 4 vs site R were not significant, and the slope showed a negative trend at all sites in 2012, the time at which restoration works occurred (Fig 4). However, the linear regressions were significant, and the slopes were positive for sites 2–4 vs the R site in 2013 one year after the restoration actions and for sites 1, 3, and 4 vs the R site in 2014 (Fig 4). The tested MLRM showed that there were not significant effect of the rainfall on the flooding duration at any sites and years (S1 Table).

### Dominant frequency of flooding levels during and after restoration

For all sampling sites the SA indicated changes in the water level at 9, 20, and 43 days in 2012 (Fig 5A). These results do not correspond to the hydrological phenomena characteristic of

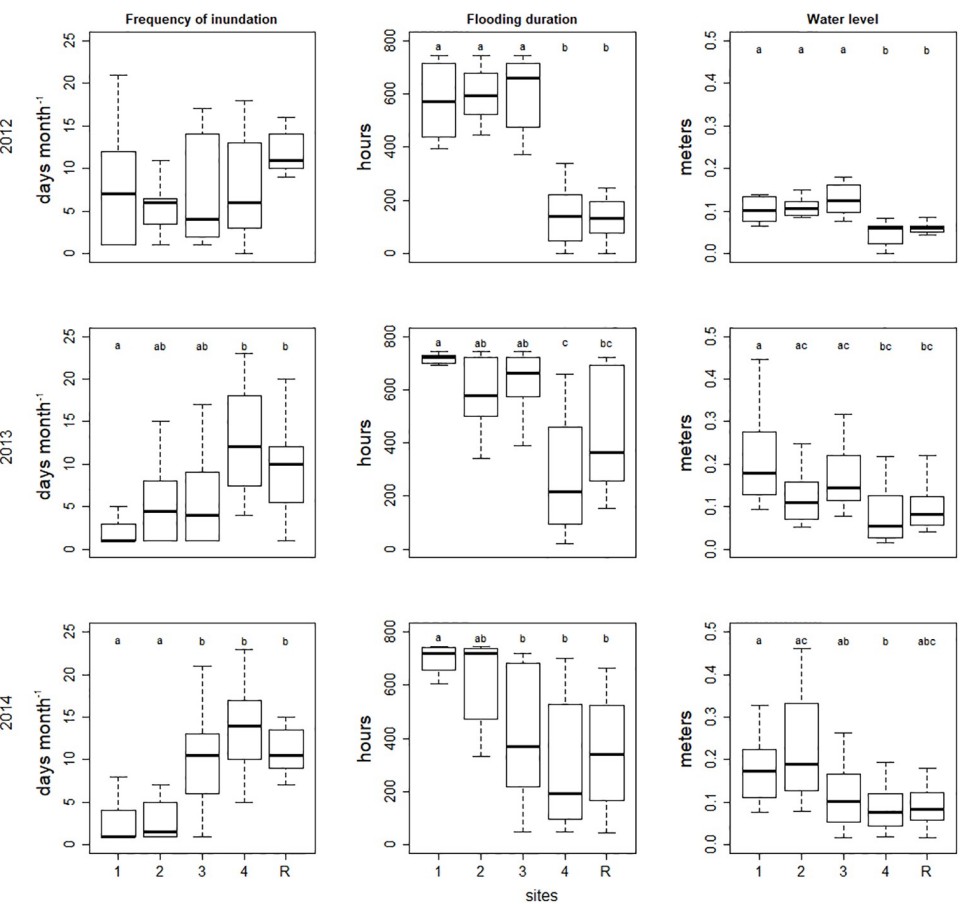

**Fig 3. Box plots indicating hydroperiod patterns (flood duration, frequency, and level) in three consecutive years after implementation of restoration activities in impaired mangrove areas in Laguna de Términos, Campeche, Mexico.** The boxplot represent the 25 to 75 percentiles (interquartile range) and the maximum—minimum distributions if the data, respectively. The line in the box is the median.

the Laguna de Términos, such as tidal and climatic seasons. However, the diurnal and semi-diurnal tidal regimes were observed in all sites (Figs 5A and 6A). After the restoration (2013), the SA for all sampling sites indicated changes in the water level as compared to 2012 with cycles of 16, 38, and 76 days. The cycle of 16 days associated with the astronomical tide (Figs 5B and 6B). In 2014, after completing the activities of hydrological and sedimentological restoration, changes in the water level related to climate seasonality were recorded in periods of 121 and 153 days for site 2 and 3 to 4, respectively (Fig 6C). Monthly changes in cycles of 28 days and the changes due to the tide were recorded in cyclesof about 15 days for sites 2, 3, 4 and R (Figs 5C and 6C). However, in site 1, the secondary channel was excavated without following the preferential flow paths, seasonality was not recorded during 2014, although there were changes in the cycles of water level (27 days) and due to the tide (15 days) (Fig 6C). These results indicate that the hydrological reconnection with the Laguna de Términos was successful.

## Biogeochemistry

The mean values of salinity before and after the restoration was compared through a two-sided t-test. Significant differences in the mean of salinity concentration were observed

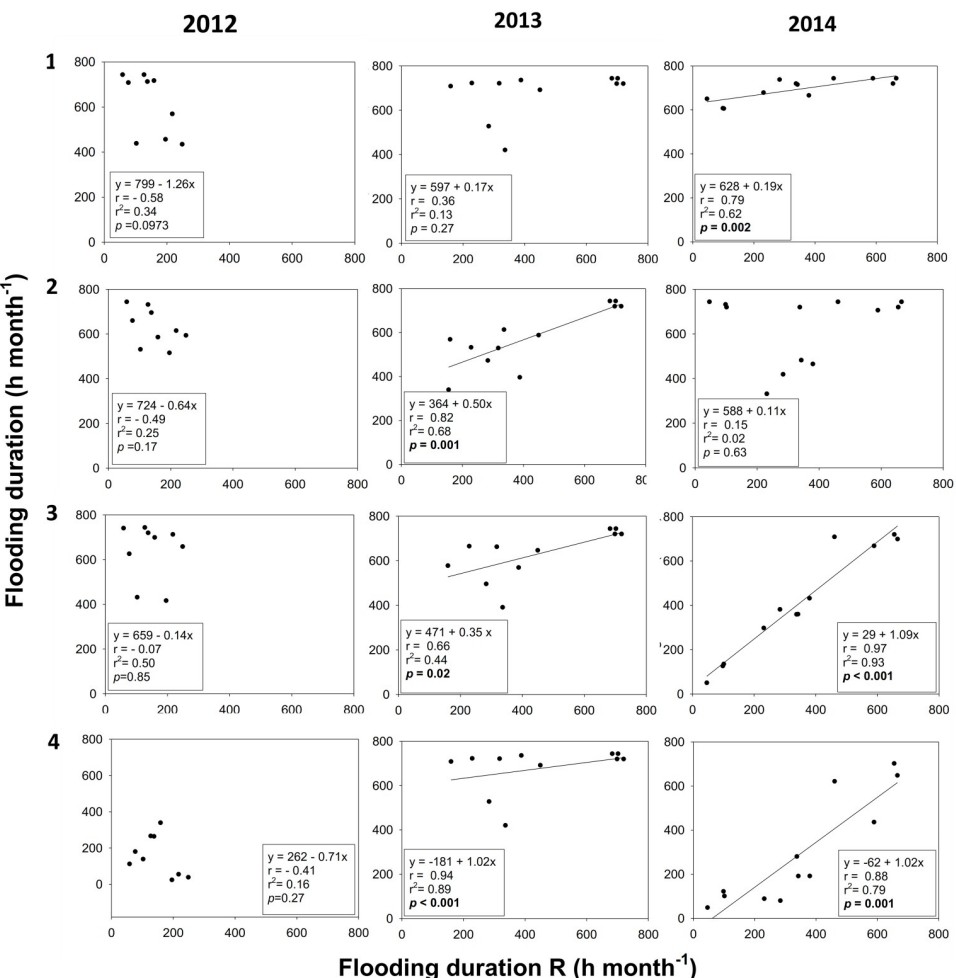

**Fig 4. Linear regressions comparing flooding duration in the impaired sites 1, 2, 3 and 4 vs. the reference site (R) in 2012 (left panels), 2013 (central panels) and 2014 (right panels).**

before and after the hydrological restoration in both, transect 1 (mean before = 78.09; mean after = 56.09; t = 15.97, df = 213.96, p < 0.0001), and 2 (mean before = 91.32; mean after = 70.92; t = 6.95, df = 75.77, p < 0.0001; Fig 7). In addition, a spatial pattern was identified in T2 (Zone II) with porewater salinity increasing from the inlet of the channel adjacent to the lagoon to the inner part of the restoration area (500 m). This pattern was present every year.

Furthermore, there were significant differences when the interaction between salinity, flooding, and sites was accounted for ($F = 4.6$, df = 52., $p = 0.01$) with higher salt concentration occurring in the periods of minimal flooding. Regarding the sulfide concentration (Fig 8), significant differences were found between flooding (minimum and maximum) and sites ($F = 7.8$, df = 52, $p = 0.001$). The highest sulfide concentrations (58 to 76 mg L$^{-1}$) were recorded only in the impaired sites during the season of maximum flooding while the lowest concentrations were found in the season of minimum flooding in the impaired (13.6 mg L$^{-1}$) and R sites (15.6 mg L$^{-1}$).

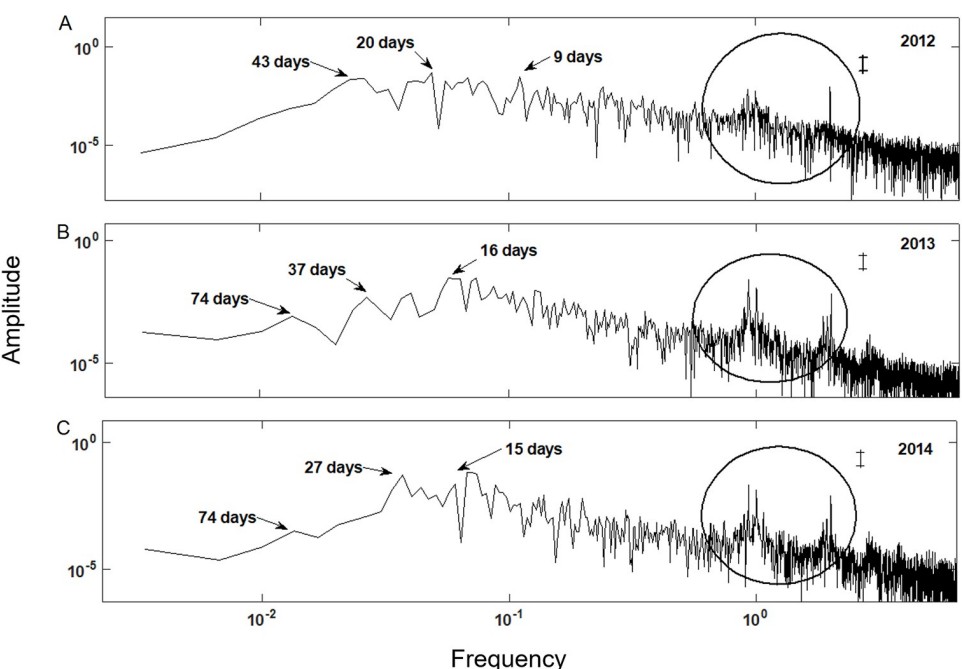

**Fig 5. Dominant frequency spectrum for the water level data of site 1 in 2012 (A), 2013 (B) and 2014 (C). The circle indicates the diurnal (left hand peak) and semidiurnal (right hand peak) tides.** The black bar indicates the confidence interval.

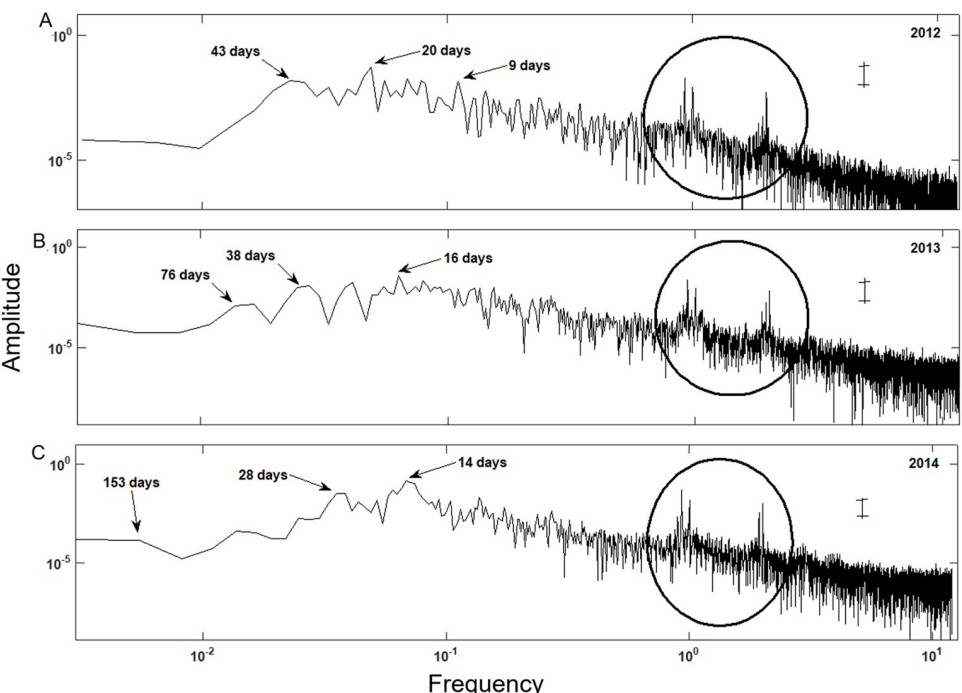

**Fig 6. Dominant frequency spectrum for the water level data of the reference site (R) in 2012 (A), 2013 (B) and 2014 (C).** The circle indicates the diurnal (left hand peak) and semidiurnal (right hand peak) tides. The black bar indicates the confidence interval.

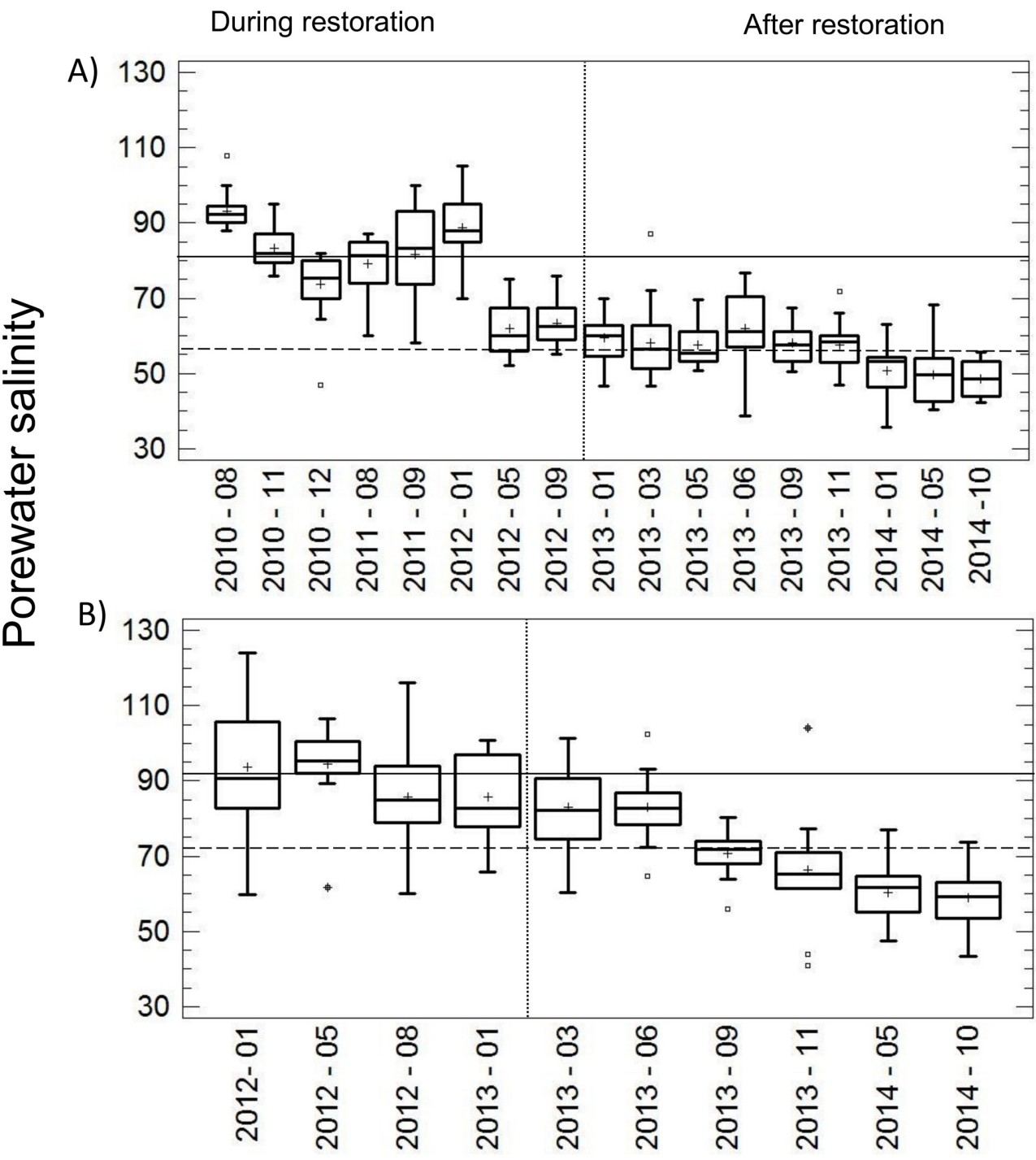

**Fig 7. The behavior of porewater salinity during and after the restoration actions is shown.** (A) Transect T1, located in Zone I where the restoration actions began in 2010 and ended in 2012 (this time-lapse considered as during the restoration actions); the years 2013–2014 were considered as after the restoration. (B) Transect T2 located in zone II: 2012 during restoration and 2013–2014 after restoration. The continuous and discontinuous lines represent the median values of the porewater salinity during and after the restoration actions, respectively. The boxes show the inter-quartile range, the thick black lines within the boxes are the medians, the cross represents the average, and the whiskers are the minimum and maximum values.

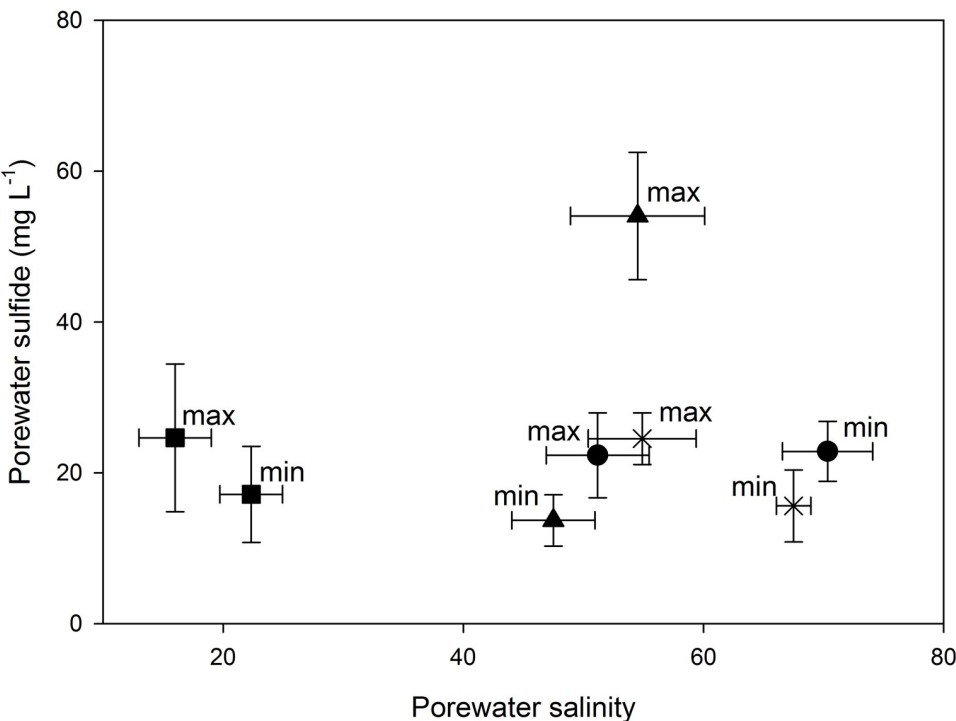

**Fig 8. Porewater sulfide vs. porewater salinity in the minimum and maximum floods in the restored site 1–4 (circle), the undisturbed reference site (cross) and other impaired sites (triangle) in Isla del Carmen, Campeche, Mexico.** The values represented by the squares are from a conserved mangrove site in South Florida, USA [59].

## Discussion

By using a microtopography-based model to identify the preferential flow paths, the entire channel was hydrologically restored, cloging was avoided, and it became the main entry point of tidal water.

We compared two approaches to rehabilitate the hydrology of mangroves; both cases involved clearing and dredging. However, in Zone I, connections between the main water body and the impaired mangrove areas were selected based only on the distance to the main tidal channel. Alternatively, in Zone II, the hydrological connection was enhanced by following the preferential flow paths inferred from the microtopography model. The limiting factor for the first approach was the selection of the appropriate tidal channels when lacking topographical information. Additionally, when excavating linear channels, the exportation of sediment to the lagoon increased, and the channel was rapidly clogged. Applying such a methodological approach require a considerably higher investment of money and time to keep the water flow moving when compared with the zig-zag shaped channels.

The results suggest that following the flow paths is more effective than merely connecting the impaired mangrove to the nearest water body. For example, the dredging of the channels of sites 1 and 2 resulted in a lack of synchronicity between the hydroperiod and dominant frequencies as compared to site R. In site 1; we did observe an increased time and level of flooding while flooding frequency decreased. However, in site 2, the time, level, and frequency of flooding decreased which resulted in more significant flooding in site 1 and draining in site 2. The ecological effect of a higher draining in site 2 was the death of mature trees, juveniles and seedlings of *A. germinans* [60]. The opening of the channel at site 1 had a severe impact on the

vegetation of site 2, supporting the reports that hydrological alteration can lead to rapid mortality of healthy and undisturbed mangroves [61].

The modification of the topography may increase the strength of the tidal ebb and flow as topographic levels are reduced [33]. In Zone II, the hydrological gradient produced by tides and by the internal circulation caused by the wind spontaneously carved secondary tidal channels <1 m in width and <0.5 m in depth. Such secondary tidal channels constituted a hydrological network connected to the broader and deeper excavated channels.

The time of flooding of the restored vs. the R site correlated with the connectivity to Laguna de Términos. On the other hand, the Fourier analysis allowed the observation of seasonal patterns with greater detail than linear regressions, enabling to associate it to tides, wind, and the water flow in Laguna de Términos. Accordingly, this is a successful strategy to explore the temporal changes in the surface and subsurface water levels [42,55].

After restoration, significant changes in the time and level of flooding occurred. Moreover, seasonal and astronomical tidal variations were detected, but only two years after conclusion of the hydrological restoration. Because the mangrove forests of Isla del Carmen are dependant on the tides and not on the freshwater supply [42, 46], no effect of the rainfall was evident for any year after the implementation of the restoration works. On the other hand, a differential seasonal effect was identified in site 2, and 3 during the study. This variation occurred because site 2 is adjacent to a patch of undisturbed mangrove, and it has relict vegetation over a higher topographic level (0.138 masl), contrasting with the impaired area where only dead trunks were found. Finally, site 1 did not present a significant correlation with site R, and fluctuations in the levels of flooding were related to the astronomical tides, but not to seasonality.

Soil subsidence occurred at the sites where dead trees were found (site 1 and 4). This phenomenon was caused by the decomposition of organic matter and dead roots during the process of degradation of the mangroves. The topographic elevation recorded at sites 1 and 4 was 0.004 and 0.031 masl, respectively, which suggests that the soil collapsed between 0.10 and 0.16 m. The collapse of the soil, in turn, increased the level and duration of flooding to values similar to those of the fringe mangrove (on the edges of the Laguna de Términos), which explains why individuals of *R. mangle* colonized these sites after the hydrological rehabilitation. These results confirm the postulate that, while the restoration of the mangroves serves to rehabilitate the sites, recovery of the original biological community remains a challenge [40,62].

Accordingly, the changes in the hydroperiod in restored mangroves determined the behavior of the biogeochemical variables. The identified salinity and sulfide patterns are associated with maximum and minimum flooding during the rainy and dry season, respectively [59]. During the maximum flooding there were lowest frequencies, highest levels and longer times of flooding, which reduced porewater salinity and oxygen concentration but increased soil sulfide concentration [63]. In the minimum flooding, there were higher frequencies, lower levels and shorter times of flooding, increasing oxygen, and decreasing sulfide concentration. In a flooding condition with null precipitation, the high temperatures increased evaporation in the impaired sites, drying the soils and chronically increasing the soil salinity and the physiological stress of the mangroves [61]. Concentrations of sulfide as high as 34 mg L$^{-1}$ are toxic to the mangroves, and salt marshes decreased aboveground and belowground production [64–66].

The effects of restoration on the hydrology and biogeochemical conditions were similar in both study zones. However, the hydrological restoration based on the preferential flow paths was much more effective in cost-benefit terms. For example, in the zone where the preferential flow paths were not applied, the cost of excavating tidal channels was up to US$10,000 ha$^{-1}$. While because preferential flow paths eliminated the need to excavate more channels of different lengths, widths, and forms to connect with the adjacent water bodies, in the zone where the preferential flow paths were used the cost was only just US$5,300 ha$^{-1}$. Through the

identification of the preferential flow paths, it is possible to know the more appropriate place to excavate the main tidal channel, which in turn helps to spontaneously create a network of channels that follows the microtopography of the terrain. This in turn, appears to be an efficient way to transport the germplasm produced in the preserved mangrove forest, reducing the costs of restoration that involve the sowing of propagules and planting of saplings which may range US$62,689 ha-[1] to US$108,828 ha-[1] [67,68]. Such a cost is 92 to 95% higher than that required when implementing the preferential flow paths.

The recovery of hydrological connectivity was also evidenced when measuring porewater salinity and sulfides during and after the actions of restoration. Specifically, in Zone I, the mean salinity decreased from 78.09 to 56.09 two years after the restoration. The decrease in salinity from 91.32 to 70.92 in Zone II, occurred only one year after excavating the tidal channels. Although soil salinity in both zones decreased after hydrological reconnection, by using the preferential flow paths helped to achieve greater efficiency in terms of the invested resources and time required. Furthermore, through the restoration of the flooding regimen and with the management (maintenance) of the channels, the survival of the mangroves is guaranteed in the long term (>50 years) [69, 70].

The changes achieved in the hydroperiod and biogeochemical characteristics allowed for a positive response in terms of the dispersion and establishment of new seedlings. These changes also allowed the seedlings' development into juveniles in the relict mangrove zones [60]. However, remaining high levels of sulfide and the collapse of the soil only favor the existence of stressed and dwarf mangrove forests. These conditions also increased the anoxic soil, the level, and the duration of flooding, and do not yet form the suitable environment for the development of a mangrove forest comparable to that of the reference site [29,71]. However, the reactivation of the process of biomass production, which is an indicator of success in the restoration of the mangroves [51], was achieved. Finally, there is evidence showing that the reactivation of the hydroperiod gave rise to ideal habitat conditions for fish, crustaceans, and birds [72].

## Conclusions

1. The hydrological restoration was based on the identification of the preferential flow of shallow water, using a micro basin approach on a digital elevation model for the first time. This procedure allowed for an informed selection of sites in which to clear, desilt, and excavate primary and secondary tidal channels.

2. The Fourier's spectral analysis helped to determine that two years after the implementation of the restoration works, the flooding levels showed seasonal and astronomical variations.

3. The hydrological restoration reactivated the hydroperiod by reconnecting the restored sites with the lagoon, which resulted in a reduction of sulfide and salt concentrations in superficial and porewater.

4. The monitoring of water levels series and the biogeochemical parameters is essential when comparing impaired vs. reference sites and when documenting the changes due to restoration activities. This process also allows for the early identification of potential hydric stress to prevent further impairing of mangrove forests.

5. The analysis and modeling of the microtopography and flow paths of shallow water in impaired mangroves and their adjacent sites significantly improve the ecological and economic efficiency effects of hydrological restoration actions.

## Supporting information

**S1 Table. The coefficients of the predictors (year and rainfall) for flooding duration at the study sites.** The last two columns show the significance of the model and the amount of variance they explain.
(DOCX)

## Acknowledgments

We thank Herminia Rejón Salazar and the 'Community of Mangrove Restorers' from Isla Aguada for their support with the field work, Tomás Zaldívar Jiménez, Mario Alejandro Gómez Ponce, Hernán Álvarez Guillén, Andrés Reda Deara and Fermín S. Castillo-Sandoval for their assistance with logistics and field data collection. To José Hernández Nava from CONANP-Laguna de Términos for the facilities to carry out our surveys.

## Author Contributions

**Conceptualization:** Rosela Pérez-Ceballos, Arturo Zaldívar-Jiménez.

**Formal analysis:** Rosela Pérez-Ceballos, Julio Canales-Delgadillo, Haydée López-Adame.

**Investigation:** Rosela Pérez-Ceballos, Arturo Zaldívar-Jiménez.

**Methodology:** Rosela Pérez-Ceballos, Arturo Zaldívar-Jiménez.

**Software:** Rosela Pérez-Ceballos, Haydée López-Adame.

**Supervision:** Rosela Pérez-Ceballos, Arturo Zaldívar-Jiménez, Jorge López-Portillo, Martín Merino-Ibarra.

**Writing – original draft:** Rosela Pérez-Ceballos, Arturo Zaldívar-Jiménez.

**Writing – review & editing:** Rosela Pérez-Ceballos, Arturo Zaldívar-Jiménez, Julio Canales-Delgadillo, Haydée López-Adame, Jorge López-Portillo, Martín Merino-Ibarra.

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
