## [Decision Letter · Decision Letter 0]

27 Aug 2019

PONE-D-19-19301

Determining hydrological flow paths to enhance restoration in impaired mangrove wetlands

PLOS ONE

Dear Mr. Zaldivar-Jimenez,

Thank you for submitting your manuscript to PLOS ONE. After careful consideration, we feel that it has merit but does not fully meet PLOS ONE’s publication criteria as it currently stands. Therefore, we invite you to submit a revised version of the manuscript that addresses the points raised during the review process.

We would appreciate receiving your revised manuscript by Oct 11 2019 11:59PM. To enhance the reproducibility of your results, we recommend that if applicable you deposit your laboratory protocols in protocols.io, where a protocol can be assigned its own identifier (DOI) such that it can be cited independently in the future. For instructions see: http://journals.plos.org/plosone/s/submission-guidelines#loc-laboratory-protocols

We look forward to receiving your revised manuscript.

Kind regards,

Rodolfo Nóbrega

Academic Editor

PLOS ONE

Journal Requirements:

4. We note that Figures 1 and 2 in your submission contain map/satellite images which may be copyrighted. All PLOS content is published under the Creative Commons Attribution License (CC BY 4.0), which means that the manuscript, images, and Supporting Information files will be freely available online, and any third party is permitted to access, download, copy, distribute, and use these materials in any way, even commercially, with proper attribution. For these reasons, we cannot publish previously copyrighted maps or satellite images created using proprietary data, such as Google software (Google Maps, Street View, and Earth). For more information, see our copyright guidelines: http://journals.plos.org/plosone/s/licenses-and-copyright.

You may seek permission from the original copyright holder of Figures 1 and 2 to publish the content specifically under the CC BY 4.0 license. 

If you are unable to obtain permission from the original copyright holder to publish these figures under the CC BY 4.0 license or if the copyright holder’s requirements are incompatible with the CC BY 4.0 license, please either i) remove the figure or ii) supply a replacement figure that complies with the CC BY 4.0 license. Please check copyright information on all replacement figures and update the figure caption with source information. If applicable, please specify in the figure caption text when a figure is similar but not identical to the original image and is therefore for illustrative purposes only.The following resources for replacing copyrighted map figures may be helpful:

5. Please include a copy of Table 1 which you refer to in your text on page 11.

Reviewers' comments:

Reviewer's Responses to Questions

**Comments to the Author**

1. Is the manuscript technically sound, and do the data support the conclusions?

Reviewer #1: Yes

Reviewer #2: Yes

2. Has the statistical analysis been performed appropriately and rigorously? 

Reviewer #1: Yes

Reviewer #2: Yes

3. Have the authors made all data underlying the findings in their manuscript fully available?

Reviewer #1: Yes

Reviewer #2: Yes

4. Is the manuscript presented in an intelligible fashion and written in standard English?

Reviewer #1: Yes

Reviewer #2: Yes

5. Review Comments to the Author

Reviewer #1: In the manuscript "Determining hydrological flow paths to enhance restoration in impaired mangrove wetlands.", the authors proposed to identify the hydrological flow paths to improve hudrological ad sedimentological restoration in an impaired mangrove forest. Local hydrological variables were used as indicators of restoration success, and monetary expenses were also considered.

Introduction provides appropriate literature that address and contextualize the main topic of the manuscript. Material and methods are well described, although some details could be improved as described below. Data presented appear to be reliably collected and appropriately analyzed. The subject is of interest to the readers of Plos One, since this research work is of interest to a wide audience. However, some issues have yet to be considered.

The text referring to the last paragraph (Lines 456-461) does not yet justify the results described in the manuscript. The changes achieved still depend on the progressive rehabilitation of local hydrology and biochemistry. However, there has been a noticeable improvement in salinity and sulfide concentrations, but still at high levels that only favor the existence of stressed and dwarf mangrove forests. The suggestion is that the discussion highlights that the conditions generated after intervention in the channels have not yet formed the suitable environment for the development of a mangrove forest comparable to that of the “R site” forest or the most developed one in the region.

Line 140: Indentation of the first line of the paragraph.

Line 151: Please, insert the species author´s name... Rhizophora mangle L. Avicennia germinans (L.) L. when species name appear for the first time.

Line 158: A. germinans instead of full name.

Line 160: A. germinans instead of full name.

Line 161: R. mangle instead of full name.

Line 163: The topic (2.2 Hydrological restoration) should be inserted in the subtopic (2.3.1 Excavation of channels).

Line 166: diameter at breast height (DBH).

Line 170: Please, observe the changes to the order of the sections numbering

Line 172: millimeter instead of mm.

Line 200: In the topic (2.4 Environmental monitoring), please, improve the text to better explain (R = reference site).

Line 201: Indentation of the first line of the paragraph.

Line 204-207: ind ha-1 instead ind.ha-1

Line 269: In the sentence: “ the data did not meet assumptions of normality.” Please, insert “and homocedasticity.”

Line 274: homocedasticity instead of homocedacity

Line285: Please, write the method used.

Line 290: 004 masl or 0.04 masl?

Line 318: There is no Fig.4B. I suppose it is Fig.4

Line 341 and 344: Please, indicate in figures 5 and 6 the meaning of A, B, and C.

Line 349: Please note indentation of the first line of the firs paragraph in this topic.

Line 354: Even if salinity values in ppt and PSU are nearly equivalent, please stick with justo ne term. In fact, salinity is a ratio, which value is dimentionless. There is no need to be follow by any unit.

Line 421: 0.004 or 0.04 masl?

Line 435: salt instead spartina.

Line 441: delete “el”.

Line 447: correct “ha-1”.

Fig. 2 – site 5 seems to be site R. If it is correct, please, indicate it in the figure legend.

Fig. 3 – “month-1” instead “month-1”

Fig. 4 – In principle, there is no rende if the regression coeficiente are not significant. So, please, remove the trend line where it is applied.

Fig. 5 – Frequency instead Frecuency.

Fig. 7 – ppt or PSU...?

Fig. 8 – mg L-1 instead mg l-1 ppt or PSU ?

Finally, I do recomend the manuscript for publication in Plos One after a review based on above general comments.

Reviewer #2: The authors describe identify the hydrological flow paths to improve the hydrological and sedimentological restoration of the Laguna de Terminos, Mexico, in an impaired mangrove forest (blockage of hydrological and sediment fluxes) through a microtopographic approach. They used as proxies of restoration success changes in the hydroperiod, flooding patterns, salinity and sulfide concentration.

It is an interesting study that adds to the literature of mangrove restoration by improvements in hydrological connectivity. I have some suggestions to make the article more clear from a scientific perspective:

1. I find that the article could benefit from a comparison with other studies performing similar assessments of the effectiveness of mangrove restoration, salinity, hydrological connectivity and dredging that the authors have ignored.

*Barreto, M.B., 2008. Diagnostics About the State of Mangroves in Venezuela: Case Studies from the National Park Morrocoy and Wildlife Refuge Cuare, in: Lieth, P.H., Sucre, D.M.G., Herzog, B. (Eds.), Mangroves and Halophytes: Restoration and Utilisation, Tasks for Vegetation Sciences. Springer Netherlands, pp. 51–64. https://doi.org/10.1007/978-1-4020-6720-4_6

*Jaramillo, F., Brown, I., Castellazzi, P., Espinosa, L., Guittard, A., Hong, S.-H., Rivera-Monroy, V.H., Wdowinski, S., 2018a. Assessment of hydrologic connectivity in an ungauged wetland with InSAR observations. Environmental Research Letters 13, 024003. https://doi.org/10.1088/1748-9326/aa9d23

*Jaramillo, F., Licero, L., Åhlen, I., Manzoni, S., Rodríguez-Rodríguez, J.A., Guittard, A., Hylin, A., Bolaños, J., Jawitz, J., Wdowinski, S., Martínez, O., Espinosa, L.F., 2018b. Effects of Hydroclimatic Change and Rehabilitation Activities on Salinity and Mangroves in the Ciénaga Grande de Santa Marta, Colombia. Wetlands 1–13. https://doi.org/10.1007/s13157-018-1024-7

2. Where is the reference point R? It is not shown on the map. Also, why do you assume that the hydroperiod is the same in that control as in the other sites? A justification is needed.

3. Where are the series of water levels? Why did you choose to do your analysis int he Fourier transform of water levels and not on the water levels themselves. It is not clear.

4. You never say what is the "biogeochemical characterization" that you focused on? and why did you analyze those in particular?

5. Why have the authors not considered that the hydrological connectivity and salinity levels have recovered because of favorable climatic conditions? How has precipitation changed from the period before to the period after the event, and during restoration? In some of the studies mentioned in 1, we see that it is difficult to attribute mangrove restoration only on dredging activities since climate conditions can become favorable afterwards during restoration because of La Nina. This would greatly improve the analysis. Ruling out the effect of climatic variability on the improvement in hydrological connectivity is necessary here.

6. Nice use of statistics in the article. However, I don't think you can attribute " an increase in the the correlation coefficient (r)" p. 236 to the restoration. I think you may not be able to compare R2 just like that. Again, more precipitation on the last year and more water in the wetland would have increased the hydrological connectivity. Also, I think you cannot compare R2 between different samples in this way. Can you check if there is a particular test to do this, apart from the Wilcox? Specially, since you have too few data points, although I agree that the relationship between flood duration in R and the sites increases in 2014.

Other issues:

L. 188 what model?

Fig. 2 - It is difficult to see the agreement between the dredged channels and the stream paths based on the DEM. IN fact, some of the channels are going across elevations? Maybe you can put both on the same figure?

6. PLOS authors have the option to publish the peer review history of their article (what does this mean?). If published, this will include your full peer review and any attached files.

Reviewer #1: No

Reviewer #2: No

---

## [Author Response · Author response to Decision Letter 0]

26 Nov 2019

Journal Requirements:

Editor: When submitting your revision, we need you to address these additional requirements. Please ensure that your manuscript meets PLOS ONE's style requirements, including those for file naming. The PLOS ONE style templates can be found at

Answer: We did make the necessary changes in the manuscript to meet the specifications given in the “Manuscript body formatting guidelines.”

Editor: In your Methods section, please provide additional information regarding the permits you obtained for the work. Please ensure you have included the full name of the authority that approved the field site access and, if no permits were required, a brief statement explaining why.

Answer: In the new version we added to the methods section the name of the institution responsible for authorizing access to the study sites. Additionally, the authorization letter is attached.

Editor: We note that you have stated that you will provide repository information for your data at acceptance. Should your manuscript be accepted for publication, we will hold it until you provide the relevant accession numbers or DOIs necessary to access your data. If you wish to make changes to your Data Availability statement, please describe these changes in your cover letter and we will update your Data Availability statement to reflect the information you provide.

Answer: We added the URL to the repository to Data Availability statement (https://onedrive.live.com/?id=root&cid=5F15A79926E1BA68). 

Editor: We note that Figures 1 and 2 in your submission contain map/satellite images which may be copyrighted. All PLOS content is published under the Creative Commons Attribution License (CC BY 4.0), which means that the manuscript, images, and Supporting Information files will be freely available online, and any third party is permitted to access, download, copy, distribute, and use these materials in any way, even commercially, with proper attribution. For these reasons, we cannot publish previously copyrighted maps or satellite images created using proprietary data, such as Google software (Google Maps, Street View, and Earth). For more information, see our copyright guidelines:

Aswer: We are making available the image licence of the used Worldview 2 to make the figures 1 and 2.

Editor: Please include a copy of Table 1 which you refer to in your text on page 11.

Answer: We removed the reference to Table 1 from the text because such a table did not exist. We have rewritten the entire paragraph to refer it to Figure 3 for a better understanding of the information.

Response to the comments of Review 1 

Review 1: The text referring to the last paragraph (Lines 456-461) does not yet justify the results described in the manuscript. The changes achieved still depend on the progressive rehabilitation of local hydrology and biochemistry. However, there has been a noticeable improvement in salinity and sulfide concentrations, but still at high levels that only favor the existence of stressed and dwarf mangrove forests. The suggestion is that the discussion highlights that the conditions generated after intervention in the channels have not yet formed the suitable environment for the development of a mangrove forest comparable to that of the “R site” forest or the most developed one in the region.

Answer: Thanks for all your comments. As suggested, we enriched the discussion by highlighting that in the study area, there are not yet suitable conditions for the optimal development of mangrove forests. We also included the proper references.

Line 140: Indentation of the first line of the paragraph.

Answer: We edited the sentences as suggested.

Line 151: Please, insert the species author´s name... Rhizophora mangle L. Avicennia germinans (L.) L. when species name appear for the first time.

Answer: As suggested, we inserted the authors’ species names.

Line 158: A. germinans instead of full name.

Answer: We edited the species’ name as suggested.

Line 160: A. germinans instead of full name.

Answer: We edited the species’ name as suggested.

Line 161: R. mangle instead of full name.

Answer: We edited the species’ name as suggested.

Line 163: The topic (2.2 Hydrological restoration) should be inserted in the subtopic (2.3.1 Excavation of channels). 

Answer: We edited the structure of the manuscript to meet the reviewer’s suggestions.

Line 166: diameter at breast height (DBH). 

Answer: We edited the DBH nomenclature.

Line 170: Please, observe the changes to the order of the sections numbering.

Answer: We did check the order of the section numbering to meet the reviewer’s suggestion.

Line 172: millimeter instead of mm.

Answer: We did substitute mm for millimeter.

Line 200: In the topic (2.4 Environmental monitoring), please, improve the text to better explain (R = reference site).

Answer: We have rewritten the paragraph, and we did include additional explanations for more clarity about the R site.

Line 201: Indentation of the first line of the paragraph.

Answer: We edited the line as suggested.

Line 204-207: ind ha-1 instead ind.ha-1.

Answer: We have corrected the nomenclature as suggested.

Line 269: In the sentence: “the data did not meet assumptions of normality.” Please, insert “and homocedasticity.”

Answer: We inserted the word “homocedasticity” in the sentence as suggested.

Line 274: homocedasticity instead of homocedacity

Answer: We corrected the spelling for homocedasticity.

Line285: Please, write the method used.

Answer: The used method is already described in the seccion 2.4.2 Soil biogeochemistry.

Line 290: 004 masl or 0.04 masl?

Answer: We have corrected the numbers. The correct numbering is 0.037 masl.

Line 318: There is no Fig.4B. I suppose it is Fig.4.

Answer: We edited the reference in the text to Fig 4.

Line 341 and 344: Please, indicate in figures 5 and 6 the meaning of A, B, and C.

Answer: As suggested, we added the explanation of the meaning of A, B, and C in the caption of figures 5 and 6.

Line 349: Please note indentation of the first line of the first paragraph in this topic.

Answer: We edited the line as suggested.

Line 354: Even if salinity values in ppt and PSU are nearly equivalent, please stick with just the term. In fact, salinity is a ratio, which value is dimensionless. There is no need to be follow by any unit. 

Answer: Thank you for this observation. We choose to eliminate the units for salinity values.

Line 421: 0.004 or 0.04 masl?

Answer: We edited the number.

Line 435: salt instead spartina.

Answer: We changed the word as suggested.

Line 441: delete “el”.

Answer: We did delete “el”.

Line 447: correct “ha-1”.

Answer: We have corrected the nomenclature as suggested.

Fig. 2 – site 5 seems to be site R. If it is correct, please, indicate it in the figure legend.

Answer: We inserted R instead of 5 in the figure.

Fig. 3 – “month-1” instead “month-1”.

Answer: We edited the number as suggested.

Fig. 4 – In principle, there is no rende if the regression coeficiente are not significant. So, please, remove the trend line where it is applied.

Answer: We removed the regressions lines from the graphics where the coefficient´s significance was less than 0.05.

Fig. 5 – Frequency instead Frecuency.

Answer: We corrected the spelling of frequency.

Fig. 7 – ppt or PSU...?

Answer: We did substitute ppt for PSU.

Fig. 8 – mg L-1 instead mg l-1 ppt or PSU ?

Answer: We corrected the nomenclature as suggested.

Response to comments of Reviewer 2.

Editor: I find that the article could benefit from a comparison with other studies performing similar assessments of the effectiveness of mangrove restoration, salinity, hydrological connectivity and dredging that the authors have ignored.

Answer: Thank you for all the comments. We reviewed the suggested articles where we did find information about that restoration of the hydrological connection through channels, as well as the maintenance of them, allows the survival of the mangrove, so we cite to:

Jaramillo, F., Licero, L., Åhlen, I., Manzoni, S., Rodríguez-Rodríguez, J.A., Guittard, A., Hylin, A., Bolaños, J., Jawitz, J., Wdowinski, S., Martínez, O., Espinosa, L.F., 2018b. Effects of Hydroclimatic Change and Rehabilitation Activities on Salinity and Mangroves in the Ciénaga Grande de Santa Marta, Colombia. Wetlands 1–13. https://doi.org/10.1007/s13157-018-1024-7.

We did observe differences (Santa Marta and Laguna de Términos) about the contributions of the water sources to the lagoons. The Cienega Grande de Santa Martha receives inputs of freshwater and sediments from the rivers, which have likely contributed to the dusting of the previously excavated channels. In Laguna de Términos most of the water contributions come from the two inlets that connect the lagoon with the Gulf of Mexico. The observed effect after the opening of the tidal channels was an increase in the ebb and flow of the tides, which allowed the appearing of a hydrological network connected to the excavated channels. Moreover, we did not record the dusting of the channels. 

The following paragraph of our manuscript explains this condition:

“The modification of the topography may increase the strength of the tidal ebb and flow as topographic levels are reduced [33]. In Zone II, the hydrological gradient produced by tides and by the internal circulation caused by the wind spontaneously carved secondary tidal channels <1 m in width and <0.5 m in depth that constituted a hydrological network connected to the wider and deeper excavated channels”.

Editor: Where is the reference point R? It is not shown on the map. Also, why do you assume that the hydroperiod is the same in that control as in the other sites? A justification is needed.

Answer: We added the point for the R site to the map. The description of the site was also edited in the text for a better understanding.

3. Where are the series of water levels? Why did you choose to do your analysis in the Fourier transform of water levels and not on the water levels themselves? It is not clear.

The Fourier transform and spectral analysis have had an essential use in surface and subsurface hydrology. Spectral methods are also recognized as a critical analysis technique in the study of geophysics, oceanography, and meteorology. We propose this type of analysis to evaluate the hydrological connectivity between the mangrove and the adjacent water body.

The series of water levels were used for spectral analysis. They were not included because the dominant frequency spectra were plotted. In such a way we were able to describe different signals such as seasonality and tide. These phenomena are not evident in the time series or the hydroperiod (level, frequency, duration) alone.

The changes in the water levels within the mangrove are modulated by the different phenomena involved in the hydrological regime of Laguna de Términos. For example, the winds, increases in the tides, temperature, and rainfall can be identified when seasonal changes are recorded. The signals of diurnal and semi-diurnal tides are among the most critical short-term variations, and only by using the spectral analysis, it is possible to identify such variations, which can be understood as hydrological connectivity.

Editor: You never say what is the "biogeochemical characterization" that you focused on? and why did you analyze those in particular?

Answer: It is correct; we define what we meant for biogeochemical characterization in the introduction. We also explained the reasons to analyze it.

Editor: Why have the authors not considered that the hydrological connectivity and salinity levels have recovered because of favorable climatic conditions? How has precipitation changed from the period before to the period after the event, and during restoration? In some of the studies mentioned in 1, we see that it is difficult to attribute mangrove restoration only on dredging activities since climate conditions can become favorable afterwards during restoration because of La Nina. This would greatly improve the analysis. Ruling out the effect of climatic variability on the improvement in hydrological connectivity is necessary here.

Answer: We include in the manuscript the monthly rainfall values for 2012, 2013 and 2014. We made multiple generalized linear models that relate precipitation with the duration of the flood to explain the influence of rainfall on hydrological connectivity. The result of this new analysis indicated that there were no significant effects of the rainfall on the flooding duration. So, in the scope of our study, the variation of the rainfall is not a phenomenon that modulates changes in water levels, neither has a contribution to hydrological connectivity. Similarly, according to NOAA (2019, published online: www.ncdc.noaa.gov/sotc/global/201413) neutral ENSO conditions did not contribute to changes in water levels during the study years.

Editor: Nice use of statistics in the article. However, I do not think you can attribute " an increase in the the correlation coefficient (r)" p. 236 to the restoration. I think you may not be able to compare R2 just like that. Again, more precipitation on the last year and more water in the wetland would have increased the hydrological connectivity. Also, I think you cannot compare R2 between different samples in this way. Can you check if there is a particular test to do this, apart from the Wilcox? Specially, since you have too few data points, although I agree that the relationship between flood duration in R and the sites increases in 2014.

Answer: As decribed in response to comment five, by using multiple linear regression models we explained the contribution of rainfall to the hydrological connectivity. This new analysis complements the Wilcoxon tests. The methodological procedure was added to the methods section.

Editor: L. 188 what model?

Answer: We corrected the sentence by including “flow path”.

Editor: Fig. 2 - It is difficult to see the agreement between the dredged channels and the stream paths based on the DEM. IN fact, some of the channels are going across elevations? Maybe you can put both on the same figure?

Answer: We edited the line representing the stream paths on the DEM for better visualization.

---

## [Decision Letter · Decision Letter 1]

27 Dec 2019

Determining hydrological flow paths to enhance restoration in impaired mangrove wetlands

PONE-D-19-19301R1

Dear Dr. Zaldivar-Jimenez,

We are pleased to inform you that your manuscript has been judged scientifically suitable for publication and will be formally accepted for publication once it complies with all outstanding technical requirements.

With kind regards,

Rodolfo Nóbrega

Academic Editor

PLOS ONE

Additional Editor Comments (optional):

Reviewers' comments:

Reviewer's Responses to Questions

**Comments to the Author**

1. If the authors have adequately addressed your comments raised in a previous round of review and you feel that this manuscript is now acceptable for publication, you may indicate that here to bypass the “Comments to the Author” section, enter your conflict of interest statement in the “Confidential to Editor” section, and submit your "Accept" recommendation.

Reviewer #1: All comments have been addressed

Reviewer #2: All comments have been addressed

2. Is the manuscript technically sound, and do the data support the conclusions?

Reviewer #1: Yes

Reviewer #2: Yes

3. Has the statistical analysis been performed appropriately and rigorously? 

Reviewer #1: Yes

Reviewer #2: Yes

4. Have the authors made all data underlying the findings in their manuscript fully available?

Reviewer #1: Yes

Reviewer #2: Yes

5. Is the manuscript presented in an intelligible fashion and written in standard English?

Reviewer #1: Yes

Reviewer #2: No

6. Review Comments to the Author

Reviewer #1: The manuscript (PONE-D-19-19301R1) was critically reviewed and authors accepted and incorporated all reviewers suggestions improving data quality and turning into a useful information for mangrove restoration. From my point of view, the manuscript meet appropriate requirements for publication in Plos One.

Reviewer #2: Thanks for the answers. The following question is still unanswered: Also, why do you assume that the hydroperiod is the same in that control as in the other sites? A justification is needed.

I would use "silting" instead of "dusting" of the channels.

7. PLOS authors have the option to publish the peer review history of their article (what does this mean?). If published, this will include your full peer review and any attached files.

Reviewer #1: No

Reviewer #2: No

---

## [Editor Report · Acceptance letter]

23 Jan 2020

PONE-D-19-19301R1 

Determining hydrological flow paths to enhance restoration in impaired mangrove wetlands 

Dear Dr. Zaldívar-Jiménez:

I am pleased to inform you that your manuscript has been deemed suitable for publication in PLOS ONE. Congratulations! Your manuscript is now with our production department. 

With kind regards,

on behalf of

Dr. Rodolfo Nóbrega 

Academic Editor

PLOS ONE